# Association between Childhood Overweight and Altered Concentrations of Circulating Amino Acids

**DOI:** 10.3390/nu16121843

**Published:** 2024-06-12

**Authors:** Jéssica de Oliveira Campos, Tafnes Laís Pereira Santos de Almeida Oliveira, Oriane Vitalis, Jéssica Gonzaga Pereira, Isabella da Costa Ribeiro Nogueira, Gabriela Carvalho Jurema Santos, Karim Chikh, Carol Gois Leandro, João Henrique da Costa-Silva, Luciano Pirola

**Affiliations:** 1Laboratory of Nutrition, Physical Activity and Phenotypic Plasticity, Academic Center of Vitória, Universidade Federal de Pernambuco, UFPE, Vitória de Santo Antão 55608-680, PE, Brazil; jessica.oliveiracampos@ufpe.br (J.d.O.C.); tafnes.lais@ufpe.br (T.L.P.S.d.A.O.); jessica.gonzaga@ufpe.br (J.G.P.); carol.leandro@ufpe.br (C.G.L.); joao.hcsilva@ufpe.br (J.H.d.C.-S.); 2Laboratory of Physical Evaluation and Signal Processing, Academic Center of Vitória, Universidade Federal de Pernambuco, UFPE, Vitória de Santo Antão 55608-680, PE, Brazil; isabella.ribeiro@ufpe.br (I.d.C.R.N.); gabriela.cjsantos@ufpe.br (G.C.J.S.); 3INSERM Unit 1060, CarMeN Laboratory, Lyon Civil Hospitals, Claude Bernard Lyon1 University, 69310 Pierre Bénite, France; oriane.vitalis@chu-lyon.fr (O.V.); karim.chikh@univ-lyon1.fr (K.C.)

**Keywords:** childhood overweight, amino acids, glycine, branched chain amino acids (BCAA), metabolomics

## Abstract

(1) Background: Dysregulated serum amino acids (AA) are known to be associated with obesity and risk of Type 2 Diabetes (T2D) in adults, and recent studies support the same notion in the pubertal age. It is, however, unknown whether childhood overweight may already display alterations of circulating AA. (2) Methods: We used liquid chromatography coupled to tandem mass spectrometry (LC-MS/MS)—targeted metabolomics to determine plasma concentrations of AA and AA-related molecules in 36 children aged 7–12 years with normal weight or overweight. Clinical and anthropometric parameters were measured. (3) Results: Overweight in children is associated with an altered AA profile, with increased branched-chain amino acids (BCAA) and decreased glycine levels, with no clinically manifested metabolic conditions. Moreover, z-BMI was positively and negatively correlated with BCAA and glycine levels, respectively, even after adjustment for age and gender. We also found a correlation between the AA profile and clinical parameters such as lipids profile and glycemia. (4) Conclusions: A pattern of low glycine, and increased BCAA is correlated to z-BMI, total cholesterol, and triglycerides in overweight but otherwise healthy children. Our data suggest that, in childhood overweight, AA disturbances may precede other clinical parameters, thus providing an early indicator for the later development of metabolic disease.

## 1. Introduction

Childhood overweight and obesity significantly predispose to metabolic disease at the adult age, indicating that preventive actions to curb obesity at a young age or preventing it altogether will have positive consequences on health later in life [1].

In 2016, a WHO-supported Commission on Ending Childhood Obesity (ECHO) set ambitious weight control/reduction targets for the young population by 2025 [2], which remain far from being met. Increased incidence of overweight and obesity in children supports the need for the determination of early-age biochemical markers that may be predictive for later manifestations of metabolic and cardiovascular diseases.

Plasma concentrations of amino acids (AAs) have been long known to be altered in adults with obesity [3]. More specifically, a distinctive pattern of elevation in branched-chain amino acids (BCAA) distinguished lean individuals from individuals with obesity and was proposed to be a contributor towards insulin resistance [4]. Mechanistically, studies in rodent models have shown that feeding an excess of BCAA lead to insulin resistance, associated to inhibitory serine/threonine phosphorylation in several players of insulin signaling, including mTOR (mammalian target of rapamycin), and IRS1 (insulin receptor substrate 1) [4]. An increase in aromatic amino acids (AAA) [5] and decreased glycine also occurs in obesity [6] and obesity-associated conditions such as nonalcoholic fatty liver disease [7], type 2 diabetes [8] and polycystic ovary syndrome [9]. These observations have also been expanded to the adolescent diabetic population [10], in which the pattern of decreased glycine and elevated BCAA and AAA was associated with impairment in insulin sensitivity and adiponectin secretion. In a randomized nutritional intervention trial aiming at achieving weight loss, a concurrent reduction of the BCAAs leucine and isoleucine was observed, as well as an improvement of insulin resistance [5].

Other metabolic indexes, such as the glutamine (Gln)/glutamic acid (Glu) ratio [11], and the proline (Pro)/citrulline (Cit) ratio [12] have been shown to be predictive of later metabolic or cardiovascular risk. Finally, the global arginine bioavailability ratio (GABR), defined as arginine/[ornithine + citrulline], is also predictive of cardiovascular risk and reflects more faithfully the occurrence of reduced NO synthetic capacity as compared with the serum of each of the amino acids concurring to its calculation [13].

Here, using a liquid chromatography mass-spectrometry-based targeted metabolic profiling approach, we investigated whether serum concentrations of proteogenic AA and AA-related molecules are associated with body weight in children, an age category in which the relationship between circulating amino acids and body weight has not been as yet extensively characterized. We unveiled associations between altered AA levels and (i) common biochemical serum parameters, including triglycerides (TG) and cholesterol, (ii) blood pressure, and (iii) adiposity measures. Overall, our data suggest that alterations of serum AA in children with overweight precede clinical manifestation of altered metabolism and may be early biomarkers for later risk of metabolic and cardiovascular disease.

## 2. Materials and Methods

### 2.1. Study Design and Participants

We performed a cross-sectional study on children between 7 and 12 years of age participating in the “Crescer com Saude” (growing healthy) program. Children were enrolled through the municipality school system of Vitória de Santo Antão, Pernambuco, Brazil. Participants were divided into two groups according to their BMI z-score: a normal weight group (NW, BMI z-score < 1, *n* = 22) and an overweight group (OW, BMI z-score > 1, *n* = 14) including overweight and obese children.

Children were selected through a process of spontaneous adherence through a non-probabilistic sampling process. Subjects presenting psychological or behavioral disturbances, use of medications, physical inabilities, or any clinical condition that could compromise food ingestion, nutritional status and/or blood pressure were excluded. We also excluded girls presenting premature menarche. All information for inclusion criteria was collected from school records and/or parents. The study was approved by the Human Research Ethics committee of the Federal University of Pernambuco (protocol number 6703946) and conducted in agreement with the declaration of Helsinki. An informed consent form was signed by the children and the children’s parents before enrolment and data collection.

### 2.2. Anthropometric and Body Composition Evaluation

Children were evaluated for body mass, height, waist circumference (WC) and body fat according to a previous protocol [14]. Body weight was measured by a digital scale balance (precision: 100 g; Omron^®^, HBF-214LA, São Paulo, Brazil) and height with a portable stadiometer (precision: 0.1 cm; Avanutri^®^, AVA-040, Rio de Janeiro, Brazil). These parameters were used to calculate the body mass index [BMI = body mass (kg)/height (m^2^)]. Standardized BMI score (z-BMI) were calculated using the freely available AnthroPlus software (World Health Organization, Geneva, Switzerland, version 1.0.3). WC was measured using a flexible steel measuring tape with a scale of 0–200 cm and precision of 1 mm (Cescorf, Porto Alegre, Brazil). The waist-to-height ratio (WHtR) was calculated by dividing the WC by the height.

Triceps and subscapular skinfolds were measured following the Lohman protocol [14] for obtain body fat percentage. Both skinfolds were measured three times on the right side of the body using a digital adipometer with a precision of 1 mm (Cescorf, Porto Alegre, Brazil). Mean skinfold values were used to estimate the percentage of body fat using the Slaugther’s formula for individuals of both genders between 8 and 18 years old [15].

### 2.3. Blood Pressure Measures

A pediatric stethoscope and aneroid sphygmomanometer (Premium^©^, Medical Instruments, Wenzhou, China), previously calibrated, with an adequate pediatric cuff size (10 to 35 cm) was used to measure systolic and diastolic blood pressure (SBP and DBP). Three consecutive measurements at two-minute intervals were taken, on three different days. The SBP was defined by the first Korotkoff sound (phase I), and the DBP by the disappearance of the Korotkoff sound (phase V) [16]. All the measures were performed following national [17] and international guidelines [16]. BP measurements represent average values obtained on the three days.

### 2.4. Collection of Blood Samples and Biochemical Analyses

The blood draw was done in the morning after 8–12 h of fasting in the children’s school precinct in an appropriate room and with a qualified person following the instructions of the Ministry of Health for collecting blood samples. Blood was collected in tubes containing heparin (biochemical analysis) and clot-activating gel (serum processing). Part of the blood was used for immediate evaluation of biochemical parameters and part of the blood was centrifuged and treated to obtain serum. Serum was collected into 1.5 mL Eppendorf tubes and stored at −80 °C prior to mass spectrometry analysis. Fasting serum glucose, total cholesterol (TC), triglycerides (TG), high-density lipoprotein (HDL), and low-density lipoprotein (LDL) were measured on a LDX Cholestech device (Cholestech, Hayward, CA, USA).

### 2.5. Metabolomics Analysis and Data Processing

The serum was analyzed by targeted metabolomics using the MxP^®^ Quant 500 kit (Biocrates Life Sciences AG, Innsbruck, Austria). Metabolomic analysis of all samples was carried out simultaneously on a single kit. The kit provides specific instructions and settings for liquid chromatography (LC) and flow injection analysis (FIA) MS/MS, operating in both positive and negative ionization modes, including instrument-specific acquisition and quantification methods. Before analysis, wash solvents and mobile phases were prepared according to manufacturer’s instructions. Serum samples, external standards, and quality control samples were loaded onto the kit’s 96-well filter plates, which are preloaded with internal standards (https://biocrates.com/mxp-quant-500-kit/, accessed on 6 June 2024). The MxP^®^ Quant 500 kit allows for the identification of the 20 proteinogenic amino acids and 30 amino acids related molecules (Listed in Appendix A Appendix A). Sample preparation (using 2 × 10 µL serum), instrument analyses, quality control measures and checks, and metabolite quantification were performed in accordance with the manufacturer’s instructions. The analysis was carried out using a Waters ACQUITY PREMIER LC system coupled to a Waters XEVO-TQXS triple quadrupole mass spectrometer equipped with an electrospray ionization (ESI) source (Waters, Saint-Quentin-en-Yvelines, France). Chromatographic separation of amino acids was achieved using a MxP Quant 500 column (Biocrates Life Sciences AG, Innsbruck, Austria). The different separation and detection parameters (mobile phase, elution gradient, column temperature, flow, source parameters) have been defined by Biocrates. Besides amino acid quantification, the MxP Quant 500 kit allowed for the quantitative analysis of further metabolites, including 25 lipids families and 14 classes of small molecules the analysis of which will be reported elsewhere. Representative chromatograms for Glycine and Leucine separation and quantification are shown in Appendix A Appendix A.

### 2.6. Data Quality and Data Selection

Quality control of metabolite concentration was performed following the instructions of the MxP^®^ Quant 500 kit. Specifically, 7 calibrators (provided by Biocrates) were run prior to the biological samples, and 3 quality control (QC) samples containing known concentrations of metabolites (provided by Biocrates) were evenly spaced between the biological samples as a quality control measure. Data were collected with MassLynx^®^ software (Waters, Milford, MA, USA, https://www.waters.com/nextgen/in/en/products/informatics-and-software/mass-spectrometry-software/masslynx-mass-spectrometry-software.html, accessed on 6 June 2024) and analyzed with Biocrates WebIDQ™ software (Biocrates Life Sciences AG, Innsbruck, Austria, https://biocrates.com/webidq/, accessed on 6 June 2024). WebIDQ™ allow the calculation of >400 quantifiable metabolism indicators. In this study, the 20 proteogenic amino acids, and 30 amino acids related molecules were analyzed. Initial exploratory data analysis was performed to detect the presence of outliers and possible inaccurate information (detection above or below the detection limit). All proteogenic AA yielded measurable values (except for cysteine whose concentrations were upper limit of quantification). Of the 30 amino acids related molecules, 18 yielded measurable values. Prior to PLS-DA (partial least squares discriminant analysis) we performed auto scaling (mean-centered and divided by the standard deviation of each variable) for data normalization (Appendix A Appendix A).

### 2.7. Statistical Analyses

The data were tested for normality and homogeneity by Kolmogorov Smirnov and Levene’s tests, respectively. Data are presented as mean +/− standard deviation of the mean (SEM). Intergroup comparisons were performed by unpaired *t*-Test for normally distributed data or Mann-Whitney test for not-normal distributions. Correlation analysis among the BMI z-Score and metabolic/clinical variables was performed using Pearson’s correlation. Partial correlation analysis was also performed for adjustment for age and gender.

Partial least-squares discriminant analysis (PLS-DA) was performed to visualize the overall difference between the groups. All analyses were performed using Excel and the freely available MetaboAnalyst version 6.0 software (https://www.metaboanalyst.ca/, accessed on 10 May 2024). Statistically significant difference was set at *p* < 0.05 and noted as described in the figure legends.

## 3. Results

### 3.1. Chromatographic Analyses

LC-MS/MS analysis on the MxP^®^ Quant 500 detection kit allowed the detection of several molecular classes, including alkaloids, amine oxides, amino acids, amino acid-related metabolites, bile acids, biogenic amines, carboxylic acids, cresols, fatty acids, hormones, indole derivatives, nucleobase-related metabolites, vitamins, and cofactors. Classes of metabolites of lipid nature were separated and detected by Flow Injection Analysis (FIA)-MS/MS method and results will be reported elsewhere.

In this study >400 metabolites were detected, and we focus here on the study of amino acids and amino acids related molecules. All proteogenic AA, with exception of cysteine, and 18 amino acid related molecules, were detected and analyzed.

### 3.2. Study Design and Participants’ Clinical Features

The study population included 14 OW (38.8%), and 22 NW (61.1%) children aged 7 to 12 years. Mean age was 9.65 ± 1.23 years old. 17 children (47.22%) were boys. A full description of the study population is presented in Table 1. While anthropometric parameters (Height, BMI, z-score, WC, Waist/Height ratio, BF) significantly differed between the NW and OW group, by virtue of the predetermined group allocation, no differences in blood pressure, glycemia or lipid profiles were observed. Hence, at the time of the study, and also considering the pre-defined inclusion criteria, no discernible clinical conditions can be attributed between the groups.

### 3.3. A Subset of Circulating AA Shows a Correlation with BMI and Is Significantly Altered in Overweight Children

Targeted metabolomics analysis allowed reliable quantification of all proteogenic AA, (except for cysteine whose concentrations were above the upper limit of quantification), and of 18 AA-related molecules (Appendix A Appendix A). The cumulative concentration of the three BCAA (Figure 1A), as well as of Ile, Leu and Val taken separately, displayed a positive correlation to z-BMI (*p* < 0.05, Appendix A Appendix A). Similarly, the Fisher Ratio, representing BCAA/AAA, positively correlated to z-BMI. On the contrary, serum Gly was inversely correlated with z-BMI (Figure 1A). Upon categorization of the study participants into NW and OW groups, amino acid profiles displayed significantly increased BCAAs (Ile, Leu and Val) and significantly decreased Gly (Figure 1B). In contrast, none of the AA-related molecules displayed differential concentrations between the NW and OW groups (Appendix A Appendix A).

### 3.4. Correlations among Clinical, Anthropometric Parameters and Amino Acids Profiles

Some amino acids and AA ratios were correlated to clinical data in a statistically significant manner. Gly was positively correlated with HDL and negatively correlated with TG while BCAA was positively correlated with TG (Figure 2). Even after adjustment for age and gender some of these correlations remained, including the correlation between BCAA and TG (Table 2). Correlations between Gln/Glu and HDL, GABR and FG; Fisher ratio and SBP, Gln/Glu and DBP were found after adjustments for age and gender (Table 2).

In agreement with the correlations between z-Score BMI and Gly and BCAA, other anthropometric parameters, such as WC, WC/Height, and body fat were also correlated with these amino acids (Figure 3A). A negative and a positive correlation was found among these anthropometric parameters and Gly and BCAA, respectively. All these correlations remained after adjustments for age and gender, except the correlation between BCAA, z-BMI and WC (Table 3). Other amino acid ratios were likewise correlated with anthropometric parameters (Table 3). WC was positively correlated with the Fisher ratio, while WC/Height was negatively correlated with the Gln/Glu ratio (Figure 3B).

### 3.5. Amino Acid Profiles Change with Age

In children aged 7–12 years old, and using data from 37 women aged 20–59 years old (De Oliveira Campos, unpublished), we compared the amino acids profiles from children versus adults, observing that the amino acids concentrations mostly increase with the age, except Glu, the only AA with higher concentration in children as compared to adults (Figure 4A). Besides the absolute differences observed in amino acid concentrations over age, overweight seems to affect the concentrations of Gly and BCAA similarly in children and adults. Thus, overweight is constantly associated with decreased Gly and increased BCAA concentrations independently of age, and despite different AA concentration ranges observed at adult age versus childhood (Figure 4B).

## 4. Discussion

The association between altered plasma AA concentrations and obesity in adults has been known for decades [3] and numerous studies in the adult population suggest a causative role of altered AA metabolic signatures leading to insulin resistance [4], diabetes risk [18] and risk for coronary artery disease [19]. Importantly, reversion of AA alterations was observed in individuals experiencing weight loss or diabetes remission [20]. While BCAA levels physiologically increase throughout adulthood, subpopulations displaying higher increase rates are associated with a higher risk of incident type 2 diabetes [21]. In this investigation, we report that children without clinically manifested metabolic conditions show an association between z-BMI and specific AA profiles.

In this study, each of the three BCAA as well as the sum of the three BCAA was positively correlated to z-BMI. Contrarily, Gly was inversely correlated to z-BMI. Specifically, children with overweight have increased serum concentrations of BCAA while Gly concentrations are decreased. Higher concentrations of BCAA have been considered as an independent risk factor for insulin resistance and associated with obesity during the childhood [14].

In a cohort of children and adolescents with obesity, baseline BCAA was positively associated with BMI z-Score but not with reduced insulin sensitivity, however, after 18 months of follow-up, BCAA became positively correlated to HOMA-IR, showing an impaired glucose metabolism [22]. Furthermore, in a cohort of >2000 adults, progressively increasing BCAA levels over 2–3 decades were associated with a higher risk of type 2 diabetes in later life as compared to study participants with constant levels of BCAA [21]. Thus, while children with overweight in this study do not present increased fasting glucose levels, the presence of higher BCAA levels, as reported here and elsewhere [22] may precede a later impairment of glucose metabolism.

Impaired BCAA levels have also been related to cardiovascular risk factors. A study on 2805 healthy adults and 2736 hypertensive patients, of both genders, observed an association between the incidence of hypertension and increased concentrations of BCAA after adjustment for confounding factors [23]. Similarly, to what was observed in adults, we found a positive correlation between BCAA, Val and the Fisher’s ratio and SBP in children, even after adjustment for age and gender.

In the Women’s Health Study, a cohort of 19,472 middle-aged healthy females, elevated BCAA was also associated with elevated TG/LDL cholesterol and lower HDL cholesterol [24], a finding that was reproduced in our study.

Oppositely to BCAA, in obesity, plasma Gly concentrations are lower and inversely correlated with BMI, and prospective studies in adults have shown that hypoglycinemia at baseline can predict the risk of developing type 2 diabetes [25]. One study with a teenage population (14–18 years old) observed that obesity led to decreased concentration of Gly. In addition, an inverse relationship between Gly and inflammatory cytokine IL-6 was found suggesting a potential link between hypoglycinemia and cardiovascular risk via an inflammatory mediator [26]. Gly depletion in obesity and insulin resistance could be the result of concurring factors, including decreased gut absorption, decreased glycine biosynthesis, and increased catabolism or urine excretion [27]. The widely observed inverse relationship between BCAA and glycine, respectively increasing and decreasing in cardiometabolic disease, may be dependent on BCAA favoring Gly use as a carbon donor in the pyruvate-alanine cycle [28], leading to Gly depletion.

Besides the altered concentrations of AA, we also found a positive correlation between some disease-predicting AA ratios and anthropometric and clinical parameters. A study performed with middle-aged men (mean age 54 years) observed that an increased Fisher ratio (BCAA/AAA) was positively associated with metabolic syndrome and negatively correlated with adiponectin concentrations and insulin sensitivity, suggesting an increasing diabetes risk [29,30]. Therefore, the positive correlation between z-BMI and Fisher Ratio in our study may also constitute an early biochemical parameter suggesting a predisposition to later metabolic disease.

In a 3-year follow-up on 1000 patients submitted to cardiac catheterization surgery, decreased global GABR, representing the arginine/[ornithine + citrulline] ratio, was linked to the occurrence of major adverse cardiovascular events (death, myocardial infarction, stroke) [13], a finding that was confirmed by later independent studies [31,32]. Our observation of a positive correlation between the GABR and fasting glucose, even after adjustment for age and gender, rather than being clinically detrimental as discussed above for adult cohorts, could be a specific and non-pathological characteristic of the young age, as arginine is an amino acid that stimulates the secretion of the growth hormone (GH) [33], a key anabolic hormone supporting children growth.

We also found a positive correlation between the Gln/Glu ratio and HDL cholesterol, suggesting that the increase of this ratio could be related to a reduced cardiovascular risk occurring during adulthood [34,35]. Moreover, we emphasized that the correlation of anthropometric parameters and BCAA, Gly, Fisher ratio, and Gln/Glu also remained even after the adjustment for age and gender (Table 3), reinforcing the notion that body weight and other anthropometric parameters may serve as early indicators of metabolic disturbance.

Finally, by comparing children AA concentrations with an adult population, we observed that most of the AA concentrations are significantly higher at the adult age, with the notable exception of glutamine (Figure 3). While previous literature showed that AA concentrations follow age-dependent trajectories during adulthood [21,36], the alterations of BCAA and Gly we observed in children with overweight are remarkable as they are paralleling adulthood changes but at a lower absolute concentration [11,37]. Overall, our data support the idea that alterations of AA levels in early youth recapitulate dysregulated metabolic profiles observed in adulthood and may serve as early indicators of the likelihood of developing metabolic diseases later in life.

Our study has some limitations as the cross-sectional study design does not allow for establishing a causal relationship between altered amino acid levels and possible future incidence of disease. Also, the study is based on a small sample of relatively homogeneous ethnicities and the stage of sexual maturation (Tanner stage) in the participants was not evaluated (except for menarche as an exclusion criterion). Finally, the assessments carried out in the present study do not allow an in-depth view at insulin metabolism and body composition, as it does not include serum insulin, homeostatic model assessment for insulin resistance (HOMA-IR), for insulin metabolism, and dual-energy X-ray absorptiometry (DEXA), for body composition. Nevertheless, we highlight that our data may contribute to understanding the relationship between AA profiles and overweight. Importantly, we introduce the idea that even at a young age (7–12 years), and in the absence of any morbidity, being overweight is associated with alterations of AA profiles widely known to increase metabolic and cardiovascular risk in adults. Our observations further support the need to develop strategies to reduce the prevalence of overweight and promote a healthy lifestyle during childhood to alleviate the incidence of metabolic disease later in life. The children, belonging to the “Crescer com Saude” (growing healthy) research and lifestyle interventions program developed by University Federal de Pernambuco (UFPE), will be followed up until adolescence and early adulthood and we may expect that this follow-up will allow to reveal, retrospectively, the potential impact of the metabolic changes we observed in the health trajectory of the study subjects.

## 5. Conclusions

Our data suggest that overweight is associated, even during childhood, with a pattern of high BCAA and low glycine serum concentrations. Moreover, z-BMI was positively correlated with BCAA and negatively correlated with glycine, even after adjustments for age and gender. In conclusion, our data describe early metabolomics alterations that, in children without any clinical condition, are associated to overweight and are possibly predictive of future metabolic disease.

## Figures and Tables

**Figure 1 nutrients-16-01843-f001:**
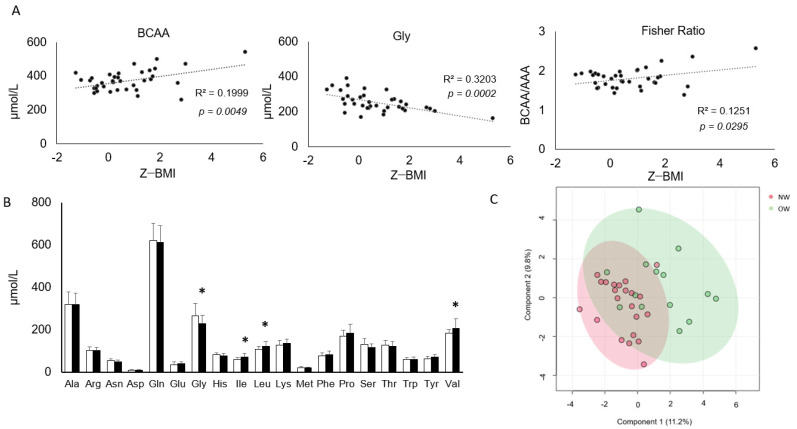
Relationships between plasma AA concentrations and BMI z-Score/weight in children aged 7–12 years old. (**A**) Pearson’s correlations between BMI z-score and branched-chain amino acids (BCAA, sum of isoleucine, leucine and valine), glycine and fisher ratio (BCAA/aromatic amino acids). R2 correlation coefficients and significance are reported on each graph. (**B**) Amino acids concentrations in normal weight (NW) and overweight (OW) children. Data represent the mean and standard deviation. Unpaired *t*-test. * *p* < 0.05. (**C**) Partial least−squares discriminant analysis (PLS−DA) of circulating amino acids in NW versus OW children. Component 1—Ile; Val; Leu; Gly; Ser; Tyr; SDMA (Symmetric dimethylarginine); kynurenine; creatinine; Glu; His; Lys; Cit; probetaine; Pro. Component 2—Ile; Val; Leu; Gly; Ser; Tyr; betaine; creatinine; SDMA; Kynurenine; His; Probetaine; Pro; Lys; Glu.

**Figure 2 nutrients-16-01843-f002:**
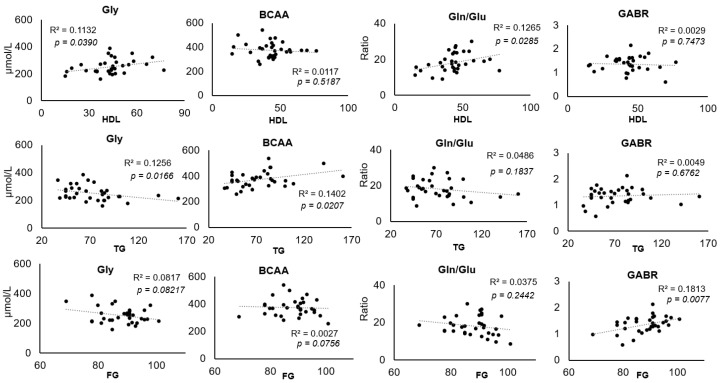
Pearson’s correlations between clinical data and amino acids in children aged 7–12 years old. Gly: Glycine; BCAA: Branched-chain amino acids; Gln/Glu: glutamine/glutamic acid ratio; GABR: arginine/(ornithine + citrulline) ratio. R2 correlation coefficients and significance are reported on each graph.

**Figure 3 nutrients-16-01843-f003:**
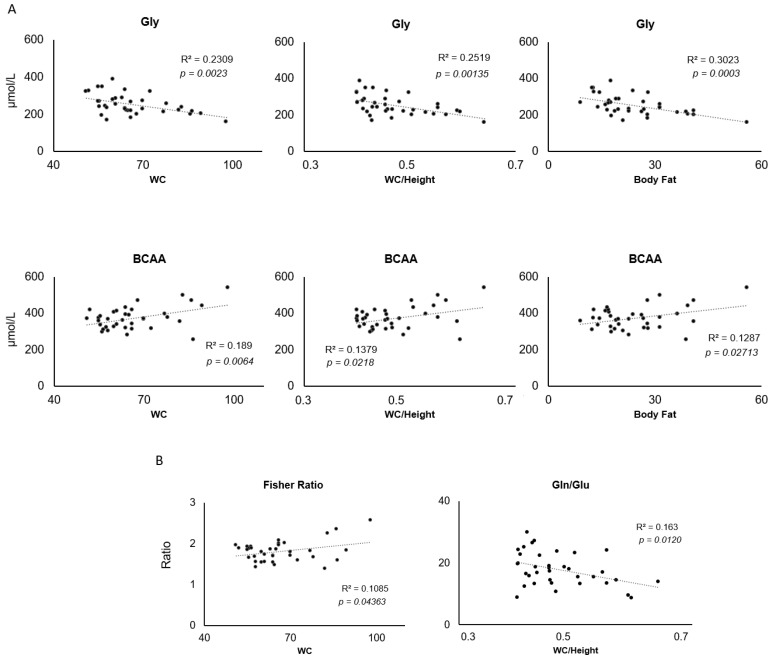
Pearson’s correlations between anthropometric measures and amino acids in children aged 7–12 years old. (**A**) Correlations between glycine, BCAA and waist circumference (WC), height/WC ratio and body fat. (**B**) Correlations between Fisher Ratio/WC and Gln/Glu; WC/height. R2 correlation coefficients and significance are reported on each graph.

**Figure 4 nutrients-16-01843-f004:**
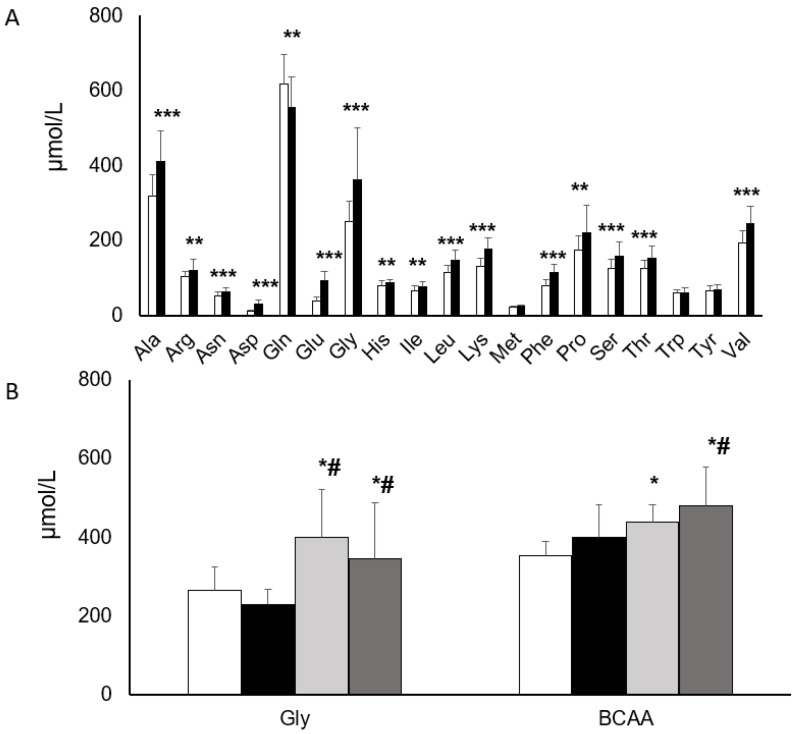
Serum AA concentrations in youth and adulthood. (**A**) Amino acids concentrations in children aged 7–12 years and adults women, aged 20–59. Data represent the mean and standard deviation. Unpaired *t*-test. ** *p* < 0.01; *** *p* < 0.001. (**B**) Concentrations of glycine and BCAA in children and women with and without overweight. Gly/Glycine; BCAA: Ile: Isoleucine; Leu: Leucine; Val: Valine; One-way Anova with Tukey post-hoc test. * *p* < 0.05 when compared to NW children; # *p* < 0.05 when compared to OW children. White bar: normal weight children; black bars: overweight children; Light gray: Normal weight women; Dark gray: Overweight women.

**Table 1 nutrients-16-01843-t001:** Characterization of anthropometric, biochemical, and blood pressure parameters of the study participants.

Parameters	Normal Weight *n* = 22	Overweight *n* = 14	*p* Value
Age ^1^	9.4 ± 1.1	9.9 ± 0.9	0.181
Height ^1^	1.38 ± 0.07	1.4 ± 0.09	0.049
BMI ^2^	16.2 ± 1.3	22.2 ± 3.8	<0.001
BMI Z-Score ^1^	−0.09 ± 0.56	2.05 ± 1.14	<0.001
WC ^2^	71.0 ± 5.18	85.0 ± 10.1	<0.001
Waist/Height ^2^	0.43 ± 0.024	0.53 ± 0.05	<0.001
BF ^2^	19.0 ± 5.7	32.4 ± 10.5	<0.001
TC ^1^	158.1 ± 32.4	152.0 ± 28.4	0.575
HDL ^1^	46.6 ± 13.1	38.7 ± 13.8	0.104
TG ^2^	64.5 ± 21.2	84.2 ± 32.4	0.050
LDL ^1^	98.4 ± 22.4	96.5 ± 21.2	0.858
FG ^1^	87.0 ± 6.99	89.5 ± 7.05	0.324
SBP ^1^	95.1 ± 6.47	98.8 ± 10.9	0.214
DBP ^1^	59.2 ± 6.76	61.5 ± 10.4	0.417

BMI: Body Mass Index, WC: Waist Circumference, BF: body Fat; FG: Fasting glucose, TC: Total cholesterol, HDL: High density lipoprotein; LDL: Low density lipoprotein; TG: Triglycerides, SBP: Systolic blood pressure, DBP: Diastolic Blood Pressure. Test *t*-student for independent samples. Statistical tests applied: ^1^ *t*-test; ^2^ Mann-Whitney test for not normal distributions.

**Table 2 nutrients-16-01843-t002:** Correlations between clinical data and amino acids and their ratios with adjustment for age and gender in children 7–12 years old.

Variables	TC		HDL		LDL		TG		FG		PAS		PAD	
	r	*p*	r	*p*	r	*p*	r	*p*	r	*p*	r	*p*	r	*p*
Glutamine	−0.012	0.948	0.399	0.026	−0.153	0.411	−0.306	0.094	−0.144	0.439	−0.007	0.97	−0.106	0.569
Glutamic acid	−0.225	0.225	−0.274	0.136	−0.126	0.499	0.126	0.5	0.254	0.168	0.222	0.23	0.393	0.029
Glycine	−0.112	0.549	0.304	0.096	−0.23	0.213	−0.293	0.11	−0.298	0.103	−0.208	0.26	−0.218	0.239
Isoleucine	0.072	0.701	−0.082	0.661	0.037	0.845	0.467	0.008	−0.088	0.639	0.289	0.115	0.198	0.285
Leucine	0.027	0.884	−0.041	0.828	−0.009	0.962	0.352	0.052	−0.094	0.614	0.147	0.431	0.073	0.697
Phenylalanine	−0.019	0.918	−0.216	0.243	0.068	0.718	0.117	0.531	−0.021	0.91	−0.194	0.295	0.046	0.807
Tryptophan	0.048	0.798	0.239	0.196	−0.053	0.777	−0.067	0.721	0.081	0.664	−0.204	0.27	−0.373	0.039
Tyrosine	0.036	0.847	0.061	0.746	−0.029	0.877	0.251	0.173	0.248	0.178	0.144	0.438	0.132	0.479
Valine	0.019	0.921	−0.232	0.208	0.063	0.738	0.442	0.013	−0.144	0.438	0.339	0.062	0.212	0.252
BCAA	0.034	0.857	−0.153	0.411	0.038	0.84	0.446	0.012	−0.125	0.502	0.287	0.117	0.177	0.34
Fisher_Ratio	0.027	0.887	−0.174	0.348	0.053	0.775	0.354	0.051	−0.254	0.168	0.409	0.022	0.229	0.216
GABR	−0.127	0.496	−0.047	0.802	−0.122	0.513	0.058	0.756	0.443	0.013	−0.298	0.104	−0.094	0.614
Gln/Glu	0.263	0.153	0.378	0.036	0.135	0.468	−0.225	0.223	−0.163	0.382	−0.168	0.367	−0.363	0.044

BCAA: Branched chain amino acids; Aromatics: Tryptophan + phenylalanine + tyrosine; Fisher ratio: BCAA/AAA; GABR: Arg/(Orn + Cit); Hcys synthesis: Hcys/Met; Ratio Pro/Cit: Proline/Citrulline; NOS activity: Citruline/arginine; Gln/Glu: Glutamine/Glutamic acid. Significant values, *p* < 0.05. Partial correlations with adjustments for age and gender.

**Table 3 nutrients-16-01843-t003:** Correlations between clinical data and amino acids and their ratios with adjustment for age and gender in children 7–12 years old.

Variables	Z-BMI		WC		RCE		%BF	
	r	*p*	r	*p*	r	*p*	r	*p*
Glutamine	−0.217	0.217	−0.165	0.352	−0.256	0.144	−0.238	0.176
Glutamic acid	0.235	0.182	0.236	0.179	0.232	0.186	0.196	0.268
Glycine	−0.570	0.000	−0.449	0.008	−0.497	0.003	−0.519	0.002
Isoleucine	0.345	0.045	0.357	0.038	0.271	0.121	0.253	0.149
Leucine	0.253	0.149	0.237	0.178	0.168	0.341	0.189	0.285
Phenylalanine	0.055	0.757	0.036	0.842	0.069	0.7	0.074	0.677
Tryptophan	−0.133	0.455	−0.181	0.306	−0.201	0.255	−0.239	0.174
Tyrosine	0.244	0.164	0.248	0.158	0.224	0.203	0.284	0.104
Valine	0.463	0.006	0.457	0.007	0.342	0.048	0.393	0.021
BCAA	0.397	0.020	0.391	0.022	0.291	0.096	0.32	0.065
Fisher_Ratio	0.362	0.035	0.372	0.030	0.257	0.143	0.284	0.103
GABR	0.097	0.587	0.114	0.521	0.124	0.484	0.284	0.103
Gln/Glu	−0.551	0.001	−0.493	0.005	−0.478	0.007	−0.413	0.021

BCAA: Branched chain amino acids; Aromatics: Tryptophan + phenylalanine + tyrosine; Fisher ratio: BCAA/AAA; GABR: Arg/(Orn + Cit); Hcys syntesis: Hcys/Met; Ratio Pro/Cit: Proline/Citrulline; NOS activity: Citruline/arginine; Gln/Glu: Glutamine/Glutamic acid. Significant values, *p* < 0.05. Partial correlations with adjustments for age and gender.

## Data Availability

The data presented in this study are available on request from the corresponding author due to ethical reasons.

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
