# Peer review of "Association between Childhood Overweight and Altered Concentrations of Circulating Amino Acids"

_nutrients, 2024, doi:10.3390/nu16121843_

Round 1

Reviewer 1 Report

Comments and Suggestions for Authors

Dear authors,

congratulations on the interesting study.

I have several recommendations:

1. The introduction must be more detailed.

2. In the Results section you should include data about the chromatographic analyses. This area of your study is not described. You should provide a reference about the validation of the method which was involved in your study or to include a description about the validation if this method is a novel one.

3. The chromatographic conditions (type of the column, mobile phase, etc.), and the reagents must be described in the Materials and Methods section.

4. A conclusion section must be included. The conclusion must be better described.

Reviewer 2 Report

Comments and Suggestions for Authors

This article by et al looks into aminoacid patterns in overweight versus normal weight children using targeted metabolomics.

They found a positive correlation between branch chained aminoacids and BCAA/AAA ratio and z-BMI, while serum Gly was inversely correlated with z-BMI. Also, they describe interesting correlations between Gly and HDL, as well as between BCAA and TG. Gly and BCAA were also corelated with anthropometric parameters.

Authors also discuss the influence of age on aminoacid concentrations and found that overweight is constantly associated with decreased Gly and increased BCAA concentrations independently of age.

The Discussions section is interesting and well written.

Some minor comments are listed below.

Abstract: please explain the LC-MS/MS acronym

Conclusions should be expanded.

Author Response

Please refer to the atatched file

Round 2

Reviewer 1 Report

Comments and Suggestions for Authors

Dear authors,

I believe that the quality of the manuscript was improved. My recommendation is- accept in present form.